# Establishment of a Challenge Model for Sheeppox Virus Infection

**DOI:** 10.3390/microorganisms8122001

**Published:** 2020-12-15

**Authors:** Janika Wolff, Sahar Abd El Rahman, Jacqueline King, Mohamed El-Beskawy, Anne Pohlmann, Martin Beer, Bernd Hoffmann

**Affiliations:** 1Institute of Diagnostic Virology, Friedrich-Loeffler-Institut, Federal Research Institute for Animal Health, Südufer 10, Insel Riems, D-17493 Greifswald, Germany; janika.wolff@fli.de (J.W.); jacqueline.king@fli.de (J.K.); anne.pohlmann@fli.de (A.P.); martin.beer@fli.de (M.B.); 2Department of Virology, Faculty of Veterinary Medicine, Mansoura University, Mansoura 35516, Egypt; sahar_virol@mans.edu.eg; 3Department of Animal Medicine, Faculty of Veterinary Medicine, Matrouh University, Matrouh 51744, Egypt; melbeskawy@gmail.com

**Keywords:** capripox, sheeppox, SPPV, field strain, pathogenesis, challenge model, nanopore sequencing, next-generation sequencing, MinION

## Abstract

Sheeppox virus (SPPV) together with goatpox virus and lumpy skin disease virus form the genus *Capripoxvirus* of the *Poxviridae* family. Due to their great economic importance and major impact on livelihood of small-scale farmers, OIE guidelines classify capripox viruses as notifiable diseases. In the present study, we examined pathogenesis of an Indian SPPV isolate and an Egyptian SPPV isolate in sheep. Three different infection routes were tested: (i) intravenous infection, (ii) intranasal infection and (iii) contact transmission between infected and naïve sheep. Clinical course, viremia and viral shedding as well as seroconversion were analyzed in order to establish a challenge model for SPPV infections that can be used in future vaccine studies. Next to in vivo characterization, both SPPV strains underwent next- and third-generation sequencing to obtain high quality full-length genomes for genetic characterization and comparison to already published SPPV sequences.

## 1. Introduction

The genus *Capripoxvirus* within the family *Poxviridae* consists of the three species *Lumpy Skin Disease virus*, *Sheeppox virus* and *Goatpox virus* [1,2]. Capripox virus-induced diseases are characterized by systemic clinical signs such as fever [3,4,5,6], depression, diarrhea, emaciation and coughing [4]. Additionally, rhinitis, conjunctivitis and excessive salivation can be observed [3]. Diseased animals develop characteristic lesions, papules and nodules in the skin [3,6,7,8], sometimes restricted to few lesions on sparsely haired regions [3] or occurring in a generalized form affecting more than 50% of the skin surface [3,4,5,6,8]. Morbidity rate as well as mortality rate of sheeppox virus (SPPV) infections are highly variable. Whereas indigenous animals show a degree of natural immunity to SPPV [9] and low morbidities (<10%) can be observed in stable enzootic situations [6], morbidity rates up to 100% in naïve populations have been described [5,10,11,12]. A similar pattern can be seen for mortality rates of SPPV: Local breeds of endemic regions show mortality rates of 5–10% [13,14,15] while naïve populations may reach up to 100% mortality [5,10,11,12]. Moreover, European breeds are known to be more susceptible to capripox virus infections in comparison to local breeds of endemic areas [9].

Transmission of SPPV occurs mainly via aerosols and direct contact [6,8,9,10,16]. Additionally, indirect transmission via contaminated objects, feed, wool and contamination of abrasions and wounds has been reported [3,8,10]. Despite the fact that potential mechanical transmission of SPPV via insect vectors could be observed experimentally [4,9,17], insects do not seem to play a major role in the epizootic of SPPV [9]. 

SPPV and goatpox virus (GTPV) rather show host preference than host specificity [3,18]. Generally, SPPV causes clinical disease in sheep and GTPV in goats [3,4,10]. However, some isolates are more pathogenic in their respective host species and induce only mild clinical signs in the other [3,7,10], whereas others uniformly cause severe clinical infection in both host species [3,19]. For affected countries, significant economic consequences are reported [3,9], and both small-scale farmers and rural communities in developing countries as well as commercial farmers in industrialized countries suffer from the occurring production losses [3,12]. Next to direct losses caused by mortalities, drop in milk production, reduced meat production and damages to hide and skin [7,9,20], trade restrictions on national level and expensive vaccination programs lead to indirect financial losses [9,12]. In addition, SPPV is listed as a potential animal bioterrorism agent [4,10]. As a result, SPPV as well as both other capripox virus species are classified as notifiable diseases under guidelines of the World Organisation for Animal Health (OIE) [21]. 

Capripox viruses are double-stranded DNA viruses [22,23] with a large genome of approximately 150 kbp [22,23,24]. Genomes are highly conserved with interspecies nucleotide identities of at least 96%, intraspecies sequence identity greater than 99%, and vaccine strains of SPPV showing 99.9% identity to their respective field strains [23]. However, capripox virus strains can be clearly differentiated by their sequences [22,23,24]. Sequencing provides an important tool for the examination of genetic relationship and characterization of viruses. Here, especially full-length sequences are of great significance. Several capripox virus isolates have been previously sequenced to generate full-length genomes [22,23,24]. By combination of the Illumina platform (next-generation sequencing; Illumina) and the MinION platform (third-generation sequencing; Oxford Nanopore Technologies, ONT), reliable, high quality full-length sequences are attained. Whereas the Illumina platform provides short, high quality reads, the MinION platform allows real-time sequencing and produces longer reads. This hybrid sequencing approach has been successfully employed for capripox viruses [25] and other large DNA viruses, e.g., African swine fever virus [26]. 

In India, SPPV was reported for the first time in Tamil Nadu, Mumbai (formerly Bombay), and United Provinces. Only few years later, in 1936, SPPV was found in Karnataka (formerly Mysore) [11]. Nowadays, SPPV is endemic in India with outbreaks occurring in almost all states [27,28]. However, some differences in the ecosystems of the country were observed. Bhanuprakash et al. analyzed outbreaks of SPPV in India over a period of 24 years. In their study, SPPV was found to be hyperendemic in the arid ecosystem and endemic in the semi-arid ecosystem. In contrast, only low endemic situations or sporadic outbreaks were detected in the coastal ecosystem [28]. Between 2005 and 2013, a total number of 3444 pox outbreaks in sheep and goats were reported by the National Institute of Veterinary Epidemiology and Disease Informatics [29]. In comparison, and although literature is sparse, few SPPV outbreaks have been reported in Egypt during the last years. For example, pox outbreaks in sheep and goats were recorded in different provinces of lower (Kafr El Sheikh, Monfia, and Dumiat Governorates) and upper (Menia Governorate) Egypt between November 2006 and October 2007 [30]. During 2014 and 2015, several outbreaks of SPPV and GTPV occurred in the Hawamdia township of the Giza Governorate [31], and a natural outbreak of SPPV in sheep was reported in the Sharkia Governorate, April 2017 [32]. 

The present study examines the pathogenesis in sheep after experimental infection with two different SPPV isolates from Surankote, India (2013) and Egypt (2018), respectively. As performed in a recent animal study [25], three inoculation routes were compared: Intravenous and intranasal inoculation as well as contact transmission between experimentally inoculated and naïve animals. Furthermore, diagnostic tools for sheeppox virus infections were validated and the suitability of both virus isolates as challenge models in future vaccine studies was analyzed. Sequencing of both virus strains utilizing the Illumina as well as the MinION platform in a hybrid assembly approach to receive high quality full-length genome sequences allowed in depth comparison with published sequences and further genetic evaluation to gain insights into the correlation of sequence and pathogenesis.

## 2. Materials and Methods 

### 2.1. Animals

Two groups of eight 3-month-old male sheep (German Blackheaded Mutton) were housed in the facilities of the Friedrich-Loeffler-Institut—Insel Riems, Germany, under biosafety level 4 (animals) conditions. All respective animal protocols were reviewed by a state ethics commission and have been approved by the competent authority (State Office for Agriculture, Food Safety and Fisheries of Mecklenburg-Vorpommern, Rostock, Germany; Ref. No. LALLF M-V/TSD/7221.3-2-004/18, approval date 15 March 2018). All sheep were in good health and no signs indicating acute infections with other pathogens were observed. At the date of inoculation, all animals were negative for antibodies against capripox viruses. 

### 2.2. Experimental Design and Collection of Samples

Sheep were inoculated with SPPV-“India/2013/Surankote” and SPPV-“Egypt/2018”, respectively. For each virus isolate, three sheep were inoculated intravenously (i.v.), another three animals were inoculated intranasally (i.n.), and two sheep were kept as in-contact (i-c) controls. Each animal regardless of the infection route received 4 mL of virus-cell-suspension: SFT-R cells (FLI cell culture collection number CCLV-RIE0043) in ZB21 (MEM (H)) + 10% FCS were seeded in a T75 cell culture flask. After incubation at 37 °C for approximately 24 h and with confluence of approximately 90%, cells were infected with 100 µL of either SPPV-“India/2013/Surankote” or SPPV-“Egypt/2018” and incubated for 7 days at 37 °C. Subsequently, infected cells were harvested via freezing at −80 °C and virus-cell-suspension was stored at −80 °C until usage. Virus titration was performed on SFT-R cells as described in the following: 100 µL SFT-R cells (approximately 30,000 cells/well) were seeded in 96 well plate format and incubated with ZB21 +10% FCS for 24 h. Log10 dilution steps of the respective virus-cell-suspension were performed in serum-free ZB21, and cells were infected with 100 µL of each dilution step. Infected cells were incubated at 37 °C for 7 days, fixed with ice-cold acetone-methanol (50:50) and subsequently stained via internal immunofluorescence protocol. Titration revealed the following titers: 10^6.3^ cell culture infectious dose_50_ (CCID_50_)/mL for SPPV-“India/2013/Surankote” and 10^6.0^ CCID_50_/_mL_ for SPPV-“Egypt/2018”. 

During the animal trial, EDTA blood for evaluation of cell-associated viremia, serum samples for examination of cell-free viremia and serological response, as well as nasal and oral swab samples for analyses of viral shedding were taken at certain time points: 0 days post infection (dpi) (before experimental inoculation), 3 dpi, 5 dpi, 7 dpi, 10 dpi, 12 dpi, 14 dpi, 17 dpi, 21 dpi and 28 dpi. In addition, body temperature and clinical reaction scores were documented daily from 0 dpi until 28 dpi. For the latter, the previously modified clinical reaction score system for examination of lumpy skin disease in cattle [33] based on Carn and Kitching [34] was used. Human endpoint was defined at a clinical score ≥10 or reaching the criteria “abandonment” in one evaluation group. During necropsy, a defined panel of organ samples (cervical lymph node (ln), mediastinal ln, mesenterial ln, liver, spleen, and lung) as well as samples of conspicuous organs and skin lesions were taken and analyzed regarding viral genome load. 

### 2.3. Molecular Analyses of Different Sample Matrices

Initially, organ samples were homogenized in serum-free medium using the TissueLyser II tissue homogenizer (QIAGEN, Hilden, Germany). For DNA extraction of all sample matrices (EDTA blood, serum, nasal swabs, oral swabs and homogenized organs), the NucleoMag Vet Kit (Macherey–Nagel, Düren, Germany) was used according to the manufacturer’s instructions and with previously described modifications of certain volumes [25] employing the KingFisher Flex System (Thermo Scientific, Darmstadt, Germany). For control of successful DNA extraction, an internal control DNA (IC2-DNA) [35] was added to each sample during the extraction process. 

Analysis of capripox virus genome loads was performed using the pan-capripox real-time qPCR assay described by Bowden et al. [4] but enhanced with a modified probe as published by Dietze et al. [36] and utilization of the PerfeCTa ToughMix (Quanta BioSciences, Gaithersburg, MD, USA).

### 2.4. Serological Examination

Serological examination was performed using the serum neutralization assay (SNT) as well as a commercially available ID Screen Capripox double antigen (DA) ELISA (ID.vet, Montpellier, France). 

The ID Screen Capripox DA ELISA was performed according to the manufacturer’s instructions. Samples with an S/P% ratio ≥30 were defined positive.

Before the SNT, serum samples were thermally inactivated (60 min, 56 °C). Triplicates of each serum samples were diluted in serum-free medium in log2 dilution series from 1:10 to 1:1280 in a 96 well plate format. 50 µL of LSDV-Neethling vaccine strain (titer 10^3.3^ CCID_50_/mL) were added to the diluted serum samples and incubated for 2 h at 37 °C. Subsequently, approximately 30,000 cells/100 µL of Madin–Darby bovine kidney cells (FLI cell culture collection number CCLV-RIE0261) were added, followed by incubation for 7 days at 37 °C. Analysis of the neutralizing titer was performed using a light microscope (Nikon Eclipse TIND-IV-100) with the method of Spearman and Kärber [37,38]. A neutralizing titer of ≥1:20 and ND_50_/_mL_ ≥14, respectively, was defined positive.

### 2.5. Sequencing of SPPV-“India/2013/Surankote” and SPPV-“Egypt/2018”

Sequencing was done as previously described [25]. Briefly, an approach combining short high-quality reads gained by Illumina sequencing and long reads by MinION Nanopore sequencing, respectively, was applied. For high-throughput Illumina sequencing (Illumina), samples were sent to the ISO17025-accredited Eurofins Genomics lab (Eurofins Genomics Germany GmbH, Ebersberg, Germany) and subsequently prepared for sequencing on the Illumina HiSeq 2500 platform according to the company’s workflow. For the MinION platform, library preparation with the Rapid Barcoding Kit (SQK-RBK004, ONT) was conducted according to the manufacturer’s instructions. This entails a two-step protocol starting with the ligation of specific barcodes after cleavage of the genomic DNA by a barcoded transposase complex, and followed by the attachment of sequencing adapters after sample pooling. An ONT MinION sequencing device (Mk1B, ONT) in combination with a MinIT (MT-001, ONT) for live basecalling and a R9.4.1 Flow Cell (FLO-MIN106D, ONT) were utilized for real-time sequencing. To receive the highest quality reads possible, high accuracy basecalling with the basecaller Guppy (v.3.2.9, ONT) was chosen for a 24-h run. After demultiplexing, quality check and trimming of the produced reads using Guppy, consensus sequence generation was conducted in an iterative mapping approach with Geneious v.11.1.5 (Biomatters, New Zealand). The acquired full-genome sequences were compared with similar full-genome sequences obtained from the INSDC. For phylogenetic analysis of the complete genomes, these sequences were aligned using MAFFT [39]. Subsequently, maximum likelihood analysis using RAxML [40], including 1000 bootstrap replicates was performed.

The respective full genome sequences obtained in this study were submitted to the INSDC under accession Nos. MW167070 (SPPV-“India/2013/Surankote”) and MW167071 (SPPV-“Egypt/2018”). 

## 3. Results

### 3.1. Clinical Signs after Experimental Infection

The first clinical sign observed in both groups was the increase in body temperature (Figure 1). For both the SPPV-“India/2013/Surankote” and the SPPV-“Egypt/2018” groups, the body temperature started to increase in the three i.v. inoculated sheep at 3 dpi. Already at 4 dpi, all i.v.-animals of the SPPV-“India/2013/Surankote” group displayed fever with a body temperature of ≥41.0 °C, and the fever lasted in all i.v.-animals until the day of removal. Thereby, the highest noticed temperature was 41.7 °C (IND-IV-09, 9 dpi). The longest period of fever ≥41.0 °C could be observed for IND-IV-10 with 7 days. In the sheep inoculated i.n., body temperature started to increase at 5 dpi, with exception of one-day fever peak of IND-IN-14 at 3 dpi. Here, all animals showed fever higher than 41.0 °C one day later at 6 dpi. As observed for the i.v. inoculated sheep, highest measured body temperature was 41.7 °C (IND-IN-12, 7 dpi). Both i.n.-animals that had to be euthanized due to welfare reasons displayed increased body temperature until day of euthanasia. Interestingly, body temperature of IND-IN-14, which survived until the end of the study, decreased to a normal range after 8 dpi, and again increased very slightly at 15 dpi and around 26 dpi. Body temperatures of i-c sheep were in a normal range until 16 dpi (IND-IC-16) and 18 dpi (IND-IC-15). Whereas IND-IC-15 showed only mild increased body temperature between 40.3 and 40.6 °C for three days (16 dpi to 18 dpi), sheep IND-IC-15 showed fever up to 41.1 °C starting at 18 dpi and lasting until euthanasia at 27 dpi (Figure 1A).

A similar pattern could be observed for the SPPV-“Egypt/2018” group. Here, fever of ≥41.0 °C was seen for EG-IV-01 at 4 dpi first, and at 5 dpi all i.v. inoculated sheep showed body temperature of ≥41.0 °C. Highest body temperature of i.v. inoculated animals was detected for EG-IV-01, which had 41.6 °C at 5 dpi. Fever of EG-IV-02 lasted until euthanasia at 14 dpi, with the longest period of body temperature ≥41.0 °C of six days (5 dpi to 10 dpi). In contrast, body temperature of EG-IV-01 and EG-IV-03 normalized over time starting at day 13 (EG-IV-03) and day 14 (EG-IV-01) pi, and reached normal range at 16 dpi (EG-IV-03) and 17 dpi (EG-IV-01), respectively. Body temperature of animals inoculated i.n. started to increase at 5 dpi (EG-IN-05, EG-IN-06) and 6 dpi (EG-IN-04), respectively. At 6 dpi, all three i.n.-sheep had a fever of over 41.0 °C, and the clearly increased body temperature lasted until day of euthanasia. Here, highest observed body temperature was also 41.6 °C (EG-IN-05, 6 dpi, 7 dpi). Body temperature of both i-c- sheep started to increase at 9 dpi (EG-IC-08) and 16 dpi (EG-IC-07), respectively. Whereas EG-IC-08 showed fever of less than 41.0 °C that lasted until euthanasia at 14 dpi, EG-IC-07 displayed fever up to 41.2 °C (19 dpi, 27 dpi) and had to be euthanized also due to ethical reasons at 27 dpi (Figure 1C).

In general, affected animals of both groups displayed ocular (Figure 2I) and nasal discharge (Figure 2A–C), abdominal breathing, and breathing sounds. Especially intranasally inoculated sheep displayed severe, partly bloody nasal discharge. In addition, sporadic skin nodules on sparsely haired regions (Figure 2D) and generalized forms of skin lesions (Figure 2C,E–I) could be observed during the animal trial. Especially axilla (Figure 2G) and groin (Figure 2H) as well as prepuce (Figure 2D–F) were affected by nodules and lesions of the skin.

In the SPPV-“India/2013/Surankote” group, first clinical signs could be seen at 5 dpi in two i.v. inoculated (IND-IV-10, IND-IV-11) and one i.n. inoculated (IND-IN-14) sheep (Figure 1B). All three animals showed slightly reduced activity as well as marginally decreased feed and water intake. In addition, sheep IND-IV-10 and IND-IV-11 displayed slight nasal discharge as well as sporadic skin nodules, leading to clinical scores of 3.5 for IND-IV-10 and IND-IV-11 and 1.5 for IND-IN-14, respectively (Figure 1B). In the following two days, all other inoculated animals also developed clinical signs characteristic for capripox virus infection. Severity of clinical infection increased over time and four of the six experimentally infected animals had to be euthanized due to ethical reasons at 10 dpi (IND-IV-10, i.v., score of 10.0; IND-IN-13, i.n., score of 10.0) and 14 dpi (IND-IV-09, i.v., score of 9.5; IND-IN-12, i.n., score of 9.5) (Figure 1B, Table 1). IND-IV-11 had to be removed from the trial at 7 dpi on reaching a clinical score of only 5.0 at 6 dpi (Table 1) as it died peracutely in the early morning of day 7 pi. In contrast to these severely affected sheep, clinical signs of IND-IN-14 remained mild to moderate with a maximum clinical score of 5.0 at 7 dpi, and this animal recovered completely from SPPV infection by the end of the study. Both i-c animals became infected with SPPV by direct contact or aerosol, showing clinical signs typical for SPPV infection starting from 16 dpi (IND-IC-16) and 17 dpi (IND-IC-15), respectively. Whereas IND-IC-16 had only a mild clinical course (highest clinical score of 3.0 at 17 dpi), IND-IC-15 became severely affected and had to be euthanized due to animal welfare reasons at 27 dpi with a score of 11.5 (Figure 1B, Table 1). 

First clinical signs of capripox virus infection in the SPPV-“Egypt/2018” group were detected at 5 dpi in all i.v. inoculated sheep, starting with slightly reduced activity, nasal discharge, abdominal breathing and sporadic skin lesions (only EG-IV-02). At 6 dpi, first clinical signs could be observed for i.n. inoculated animals EG-IN-04 and EG-IN-06, and clinical signs of EG-IN-05 started at 7 dpi. Whereas two out of the three i.v. inoculated sheep displayed a mild to moderate clinical course (maximum clinical score of 4 (EG-IV-01) and 5 (EG-IV-03), respectively) and recovered completely from the infection, EG-IV-02 was severely affected and had to be euthanized at 14 dpi (Figure 1D, Table 2). Clinical course of the i.n. inoculated sheep at first seemed to be mild, but rapidly worsened. All three animals had to be removed from the trial due to ethical reasons (EG-IN-04, 16 dpi, score of 9.0; EG-IN-05, 16 dpi, score of 8.0, EG-IN-06, 14 dpi, score of 8.0) (Figure 1D, Table 2). In this group, both i-c sheep were also severely affected by SPPV via direct or aerosol transmission but with clear differences in time point. EG-IC-08 developed first clinical signs typical for SPPV at 9 dpi that later became very severe and thus had to be euthanized at 14 dpi with a clinical score of 8.5. In contrast, EG-IC-07 showed the first clinical signs at 16 dpi. Here, the clinical course became severe at 25 dpi, leading to euthanasia at 27 dpi with a clinical score of 10.5 (Figure 1D, Table 2).

Since the utilized clinical score system successfully used for characterization of lumpy skin disease virus (LSDV) infections in cattle [33,41] and GTPV in goats [25] was insufficient for examination of SPPV in sheep, we developed a new modified version (Table 3). The newly established score system is based on both the clinical scoring for LSDV in cattle of Carn and Kitching (1995) [34] and our previously modified version [33], considering the experiences obtained in the present study.

A clinical score of 0 displays no clinical reaction. A summarized clinical score of 1–4 characterizes a mild clinical course, whereas a summarized clinical score of 5–8 represents a moderate clinical infection. Furthermore, a clinical score of 9–12 describes severe clinical infection, and at a summarized clinical score of more than 12, the respective animal has to be euthanized due to ethical reasons.

### 3.2. Viremia and Shedding of Virus

Viremia in the SPPV-“India/2013/Surankote” group was first detected in the EDTA blood (cell-associated viremia) at 3 dpi in two sheep both inoculated i.v. (IND-IV-09, IND-IV-10) and in the serum (cell-free viremia) of IND-IV-09 but with relatively high Cq-values between 34.3 (IND-IV-09, EDTA blood) and 38.3 (IND-IV-09, serum). At 5 dpi, five out of six animals scored positive in the EDTA blood, except IND-IN-14 that showed the first positive result in the EDTA blood at 7 dpi. In the following days, lower Cq-values could be observed in the EDTA blood of the infected animals, representing viremia. Here, Cq-values from 27.1 (IND-IN-13, i.n., 10 dpi), 28.6 and 29.1 (IND-IV-09, i.v., 12 dpi and 10 dpi), to around 30 to 37.3 (IND-IN-14, i.n., 10 dpi) could be detected. In contrast to IND-IC-16 (i-c), which did not show any viremia at all, EDTA blood of i-c sheep IND-IC-15 scored positive for viral DNA at 12 dpi (Cq 37.3) and again from 21 dpi on (Cq-values around 30) (Figure 3A). Compared to the results of the EDTA blood, serum samples scored positive at fewer times. Beginning with IND-IV-09 (i.v.) at 3 dpi and IND-IN-13 (i.n.) at 5 dpi with Cq-values around 37–38, serum samples of four out of six inoculated and one of the two i-c animals scored positive over the time. Solely IND-IV-11 (i.v.), IND-IN-14 (i.n.) and IND-IC-16 (i-c) did not show any viral DNA in serum samples. Moreover, Cq-values of serum samples were consistently higher compared to those of EDTA blood. In detail, Cq-values of serum samples ranged from 32.4 (IND-IN-13, i.n., 10 dpi) to 38.3 (IND-IV-09, i.v., 3 dpi) (Figure 3B).

In the SPPV-“Egypt” inoculated group, cell-associated as well as cell-free viremia also started around 3 dpi (EG-IV-03, i.v., EDTA blood, Cq 31.8; serum, Cq 31.7). At 5 dpi, EDTA blood of five out of six experimentally infected sheep scored positive for capripox virus genome. Solely for EG-IN-05, first positive EDTA blood was detected as late as 10 dpi (Cq 35.5). At 10 dpi (EG-IC-08) and 17 dpi (EG-IC-07), positive results of EDTA blood of both i-c animals could be observed. Like in the other infection group, Cq-values decreased as the disease progressed. Lowest Cq-values could be detected for EG-IV-02 (i.v., from 7 dpi until 14 dpi, Cq-values between 30.4 and 27.6), EG-IC-07 (i-c, 21 dpi, Cq 28.0) and EG-IC-08 (i-c, 10 dpi to 14 dpi, Cq-values between 30.1 and 28.1). Remaining positive EDTA blood samples showed Cq-values around 32 to 37 (Figure 3C). First positive serum sample could be observed at 3 dpi (EG-IV-03, i.v., Cq 31.7), followed by EG-IV-02 at 5 dpi (i.v., Cq 36.9) and EG-IV-01 (i.v., Cq 35.7) and EG-IC-08 (i-c, Cq 35.3) at 10 dpi. Lowest Cq-value in serum samples was achieved for EG-IV-02 (i.v.) at day of euthanasia (14 dpi) with a Cq-value of 29.2. In general, Cq-values observed for serum samples were around 34.2 (EG-IC-08, i-c, 14 dpi) and 37.9 (EG-IV-03, i.n., 10 dpi) (Figure 3D).

In the group inoculated with SPPV-“India/2013/Surankote”, shedding of viral DNA was detected as early as 3 dpi in the nasal swabs of all three animals inoculated i.n (IND-IN-12, IND-IN-13, IND-IN-14) (Figure 4A). In addition, viral shedding via oral fluid could be observed in these animals starting at 5 dpi (Figure 4B). Hereinafter, all i.n.-animals remained positive in all taken swab samples until euthanasia (IND-IN-12, IND-IN-13) or the end of the study (IND-IN-14), respectively. In the i.v. inoculated sheep, viral shedding started marginally later compared to i.n. inoculated animals. However, at 5 dpi, nasal swabs of two out of three i.v.-animals (IND-IV-10, IND-IV-11) as well as the oral swab of IND-IV-11 scored positive for viral DNA, and IND-IV-09 displayed the first positive swab samples at 7 dpi. Here, swap samples also remained positive until the day of euthanasia, solely IND-IV-11 showed a negative oral swab sample at the day of removal. Interestingly, i-c animal IND-IC-16 showed the first positive swab sample earlier at 5 dpi (nasal swab, Cq 37.5), and swab samples of i-c sheep IND-IC-15 scored positive for capripox virus genome at 10 dpi and stayed positive until euthanasia at 27 dpi. Contrarily, no viral genome could be detected in both swab samples of IND-IC-16 at 28 dpi. Taken together, observed Cq-values were very low. For the nasal swabs, Cq-values ranged from 13.3 (IND-IN-13, i.n., 10 dpi) through around 25 to 37.5 (IND-IC-16, i-c, 5 dpi). In comparison, oral swab samples displayed higher Cq-values, ranging from 23.8 (IND-IN-12, i.n., 12 dpi) through 28 to 36.2 (IND-IC-16, i-c, 14 dpi) (Figure 4A,B). 

First positive nasal swab samples of the SPPV-“Egypt/2018” group were observed at 3 dpi. In addition to the nasal swab samples of the three i.n. inoculated sheep of the group (EG-IN-04, EG-IN-05, EG-IN-06), nasal swab sample of i.v. animal EG-IV-02 scored positive with a Cq-value of 32.8 comparable to those of EG-IN-04 (27.2), EG-IN-05 (28.8) and EG-IN-06 (34.2). At 7 dpi, nasal swab samples of both other i.v. inoculated sheep (EG-IV-01, Cq 34.3; EG-IV-03, Cq 32.1) as well as of i-c sheep EG-IC-07 gave positive results for capripox virus genome. The remaining sheep, EG-IC-08 (i-c), displayed positive qPCR results for nasal swab samples starting at 10 dpi (Cq 30.9). Cq-values of nasal swab samples ranged from 16.1 (EG-IN-05, i.n., 12 dpi) through 20 and 28 to 34.3 (EG-IV-01, i.v., 7 dpi; EG-IC-07, i-c, 10 dpi) (Figure 4C). Oral swab samples scored positive for the first time at 5 dpi for all three i.n.-sheep, and at 7 dpi (EG-IV-02, EG-IV-03) and 10 dpi (EG-IV-01) for the i.v. inoculated animals. At 10 dpi and 12 dpi, respectively, oral swab samples of EG-IC-07 and EG-IC-08 scored positive. Here, Cq-values between 23.8 (EG-IC-07, i-c, 27 dpi) and 37.1 (EG-IC-07, i-c, 10 dpi) could be observed (Figure 4D). In this group, all swab samples remained positive for viral DNA until day of euthanasia of the respective animal or until the end of the study (Figure 4C,D).

Generally, in the i.v. inoculated sheep, viremia started before the onset of viral shedding via nasal or oral fluid. However, i.n. inoculated animals displayed nasal and oral viral shedding earlier than viremia could be detected. Except EG-IC-08, which showed beginning of both viremia and viral shedding on the same day (10 dpi), for i-c animals viral shedding could also be observed earlier than viremia. Moreover, nasal swabs scored positive earlier than oral swab samples and displayed lower Cq-values. In addition, with the exception of EG-IN-04 and EG-IN-05, all animals that had to be removed from the trial before the end of the study due to severe clinical course scored positive in all four tested matrices at least on the day of euthanasia (Figure 3 and Figure 4, Appendix A).

### 3.3. Presence of Viral Gnome in Organ Samples

For each animal, a defined organ panel (certain lymph nodes: cervical, mediastinal, mesenterial as well as liver, spleen and lung) was taken during dissection. In addition, suspicious organs and several skin lesions of different stages and of various parts of the body were taken. 

In SPPV-“India/2013/Surankote” infected sheep, cervical lymph node and lung tested positive in the pan-capripox real-time qPCR in all six and five out of these six animals that had to be removed from the trial before end of the study, respectively, irrespective of the inoculation route. Comparable Cq-values could be observed compared to other taken organs, which scored positive in less animals but at least once. These organs were mediastinal lymph node (IND-IV-11, IND-IC-15), mesenterial lymph node (IND-IC-15), liver (IND-IV-11, IND-IN-13, IND-IC-15) and spleen (IND-IV-11, IND-IN-13). However, clearly lower Cq-values could be observed in all taken skin samples. Here, Cq-values from 16.6 (IND-IV-09, skin lesion of axilla) through 20 and 25 to 37.2 (IND-IN-12, skin lesion of axilla) could be seen. Both animals that survived until the end of the animal trial (IND-IN-14, IND-IC-16) were tested negative in all taken organ samples indicating complete recovery from SPPV infection (Table 4, Appendix A). 

Pattern of positive organ samples of sheep experimentally infected with SPPV-“Egypt/2018” is slightly different compared to the other infection group. Here, especially lung and spleen samples provided a great tool for post-mortem detection of capripox virus genome, which scored positive in seven out of eight sheep. Lymph node samples showed inconsistent results. Whereas four out of eight animals (EG-IV-02, EG-IN-04, EG-IC-07, EG-IC-08) displayed two or three positive lymph node samples, only a single lymph node tested positive in two sheep (EG-IV-01, S05), and lymph nodes of EG-IV-03 and EG-IN-06 were completely negative for capripox DNA. In addition, different lymph nodes scored positive. One sheep (EG-IC-08) showed motor disorders during the study, which is why three different regions of the brain were tested for capripox viral genome. However, all taken brain samples were negative, indicating no influence of SPPV infection on motoric function. For two intranasally inoculated animals (EG-IN-04, EG-IN-05), nasal septum was taken, and scored highly positive in the pan-capripox real-time qPCR (Cq-values of 21.3 and 28.1, respectively). As seen for the other group, highest viral genome load could be detected in skin samples, irrespective of the body part and stage of healing. Here, Cq-values ranging from 13.8 (EG-IV-03, encrusted skin lesion) through 18 and 25 to 32.1 (EG-IV-01, healed skin lesion) could be observed. In contrast to those animals of the other infection group that survived until the end of the study, both surviving sheep of the SPPV-“Egypt/2018” group (EG-IV-01, EG-IV-03) tested positive for capripox virus DNA in two regularly taken organ samples, but with Cq-values comparably high around 37 (Table 5, Appendix A).

As observed for viral genome load in periodically taken samples, no marked differences between both virus isolates could be observed regarding capripox virus genome in organ samples. In general, all animals except one (EG-IV-01) that showed positive results in at least one organ sample scored positive for capripox viral genome in the cervical lymph node or the lung or both, making these sample types useful matrices for post-mortem detection of SPPV infections. However, highest viral genome loads were detected in skin samples of nodules, papules and crusts, irrespective of the sampled body part (Table 4 and Table 5, Appendix A).

### 3.4. Serological Response

In the SPPV-“India/2013/Surankote” inoculated group, positive results could be observed in the DA ELISA as well as in the SNT, but with clear differences in comparison. At 10 dpi, ELISA of IND-IV-09 (i.v.) scored positive, but turned negative again. In contrast, ELISA of IND-IN-12 (i.n., 12 dpi), IND-IN-13 (i.n., 10 dpi) and IND-IN-14 (i.n., 17 dpi) gave positive results and remained positive until day of euthanasia (IND-IN-12, IND-IN-13) and the end of the study (IND-IN-14), respectively (Figure 5A). However, the SNT was positive for IND-IC-15 (i-c) at 27 dpi and IND-IN-14 (i.n.) at 21 dpi and 28 dpi (Figure 5B). The latter was the only sheep of this group that showed positive results in both serological assays. Three sheep (IND-IV-10, i.v.; IND-IV-11, i.v.; IND-IC-16, i-c) stayed negative in both tests during the animal trial (Figure 5A,B). 

ELISA results of sheep inoculated with SPPV-“Egypt/2018” were mostly not consistent, showing a similar pattern as observed for IND-IV-09. First positive results in this test were observed as early as day 7 pi (EG-IV-02, i.v.). In general, five (EG-IV-01, i.v.; EG-IV-02, i.v.; EG-IV-03, i.v.; EG-IN-05, i.n.; EG-IC-07, i-c) out of eight sheep scored positive in the DA ELISA at certain time points. Thereof, ELISA of three animals (EG-IV-01, EG-IV-02, EG-IN-05) turned negative again and in case of EG-IV-01 positive for a second time (Figure 5C). In the SNT, the first positive result could be observed at 14 dpi (EG-IV-01, i.v.), followed by 16 dpi (EG-IN-04, i.n.; EG-IN-05, i.n.) and 21 dpi (EG-IV-03, i.v.) (Figure 5D). Whereas three of these four sheep additionally scored positive in the DA ELISA, serum of EG-IN-04 (i.n.) was only positive in the SNT. In this group, two (EG-IN-06, i.n.; EG-IC-08, i-c) animals were negative in both serological tests at all tested time points (Figure 5C,D).

Generally, the used ELISA proved to be positive at earlier time points compared to the SNT but turned negative again in some cases, providing inconsistent results. In contrast, the SNT scored positive at later time points but stayed positive for the rest of the study (Figure 5).

### 3.5. Full Genome Sequencing of SPPV-“India/2013/Surankote” and SPPV-“Egypt/2018”

All collected MinION and Illumina data were used in an iterative hybrid assembly and mapping approach for generation of full-length, high quality consensus sequences. For SPPV- “India/2013/Surankote” 232,891 reads (1.6% of total 469,061 MinION and 14,486,856 Illumina reads) were assembled to a 150,369 nt consensus sequence with a mean coverage of 829-fold. Mean length of the assembled reads was 518 nt with a maximum of 17,550 nt. Long reads supporting correct assembly were achieved by MinION nanopore sequencing. For SPPV-“Egypt/2018” 509,779 reads (7.1% of total 283,977 MinION and 6,949,206 Illumina reads) were assembled to a 150,478 nt consensus sequence reaching a mean coverage of 595-fold. Mean read length was 173 nt with the longest assembled read 6147 nt in length. Sequences were compared with SPPV genomes obtained from the INSDC (Figure 6).

## 4. Discussion

### 4.1. Genetic Relationship of SPPV-“India/2013/Surankote” and SPPV-“Egypt/2018”

Full-genome sequencing of SPPV can aid in identification of pathogenicity of strains. Here, inspection of the genome and comparison of key proteins indicating possible attenuated or vaccine strains [24] revealed that neither the poxvirus B22R superfamily protein nor kelch-like repeat proteins or ankyrin repeat proteins were truncated in both SPPV strains (SPPV-“India/2013/Surankote”, SPPV-“Egypt/2018”). This is in line with the experimental finding showing clear clinical signs in the animal trial presented here. Overall similarities between available strains are high (98.8–99.9% identity) and phylogenetic studies showed that SPPV-“India/2013/Surankote” is related to a vaccine virus first described 1994 from Kazakhstan [23] and SPPV-“Egypt/2018” is similar to vaccine viruses from Turkey [23,24] (Figure 6). This points to the fact that a clear judgment on the pathogenicity on overall similarities or the phylogenetic relationship alone is not suitable.

### 4.2. Pathogenesis in Sheep after Experimental Infection

Animals became infected with SPPV, independent of the used virus isolate and infection route. Clinical (Figure 1) as well as molecular (Figure 3 and Figure 4) and serological (Figure 5) data are comparable between both groups, and transmission of virus between infected and naïve animals via direct contact or aerosols was observed. Since these are known main transmission routes for SPPV and GTPV [3,4,9,12,16], this is not unexpected. Infection started with onset of fever around 3 dpi in the i.v. inoculated animals and around 5 dpi in sheep inoculated i.n, which is in line with results previously described for experimental infection of sheep with SPPV [4,5,42,43]. Start of clinical infection of i-c animals can be seen nicely by body temperature data, and beginning of infection varied between 8 dpi (EG-IC-08, SPPV-“Egypt/2018”) and 18 dpi (IND-IC-15, SPPV-“India/2013/Surankote”). During our study, maximum body temperature of 41.7 °C was detected, as previously reported by Hamdi et al. after challenge infection of unvaccinated sheep with Turkish virulent SPPV Held strain [43]. In the present study, fever of ≥41.0 °C was observed over a period of 7 days (SPPV-“India/2013/Surankote”) and 6 days (SPPV-“Egypt/2018”), respectively, with longest period of increased body temperature (≥40.0 °C) around 12–13 days (Figure 1A,C). In literature, fever periods of 4–5 days [4], 6 days [43] and 6–10 days, depending on the virus isolate [42], have been described after experimental infection of sheep with virulent SPPV isolates. Since only two animals of each infection group of our animal trial recovered from infection and the others had to be euthanized due to severe clinical course (Figure 1B,D), an even longer period of fever after natural infection could not be excluded. Nevertheless, although most sheep during our study displayed a fever for these long periods of time, general condition, feed and water intake as well as interest in other animals of the herd and in human stuff remained rather unremarkable until beginning of clinical signs like nasal discharge, abdominal breathing or development of skin lesions. Clinical signs observed during the presented animal trial (Figure 2) are typical for SPPV infections in sheep. First, reduced activity and decreased appetite could be observed, and animals displayed nasal discharge, slight breathing sounds and marginal abdominal breathing. In addition, sporadic skin nodules were detected. These signs started around 5–7 dpi in all inoculated sheep, consistent with previous observations describing onset of clinical signs around 5 dpi [42]. Later on, generalized skin nodules and papules as well as severe respiratory problems were recorded in almost all sheep, characteristic for infections with virulent SPPV strains [4,5,7,42,43]. 

Viremia was detected at 3 dpi for the first time in i.v. inoculated sheep, and viremia of animals inoculated i.n started around 5 to 10 dpi (Figure 3). The latter is consistent with previous studies of Bowden et al. and Babiuk et al. after intradermal inoculation of sheep with different capripox virus isolates. Babiuk et al. found first positive results in EDTA blood of sheep experimentally infected with a Yemen capripox virus isolate starting 8 dpi [7]. Furthermore, Bowden et al. also detected capripox DNA in blood samples starting at 6 dpi in sheep experimentally infected with a Nigerian SPPV strain [4]. Likewise seen in the studies by Bowden et al. [4], peak of viremia of the animals inoculated i.v. or i.n. was around 10 dpi and 12 dpi in our study (Figure 3). In comparison to viremia, start of viral shedding was reversed. Here, first positive nasal swab samples could be observed at 3 dpi in i.n. inoculated animals, whereas i.v. inoculated sheep started to shed virus from 5 dpi and 7 dpi on, with one single exception at 3 dpi (EG-IV-02). Oral swab samples turned positive only few days later than nasal swab samples, around 5–10 dpi (Figure 4). Our results fit nicely to already described data of viral shedding in diverse swab samples. Previous studies describe first detection of viral genomes around 6–12 dpi [4,7]. In our study, nasal swabs turned out to be more sensitive than oral swab samples (Figure 4), which had already been described for LSDV in cattle [41] and GTPV in goats [25]. The described differences in the start of viremia and viral shedding can be explained by the different inoculation methods used. Whereas first replication of virus after i.v. inoculation appears in the blood leading to early viremia, i.n. inoculation is followed by local replication of virus in the nasal mucosa. Since i-c animals of our study showed viral shedding earlier than viremia like it was observed for sheep inoculated i.n. (Figure 3 and Figure 4) and in some cases no viremia could be detected at all (Figure 3), i.n. inoculation seems to reflect natural infection better than i.v. inoculation. Similar phenomena could be also observed in goats experimentally infected with Indian GTPV [25]. 

As shown before [4,7,42], highest viral genome loads could be detected in skin lesions of affected sheep during our study, confirming skin samples as very useful sample types for diagnosis of SPPV infections in sheep. In addition, lung samples scored positive in the pan-capripox real-time qPCR during the present study indicating robust replication of SPPV in the respiratory tract (Table 4 and Table 5), which has been reported previously [4]. Therefore, lung samples also provide a reliable sample type for post-mortem detection of SPPV infections in sheep. Furthermore, Hajjou et al. detected high viral loads in the lymph nodes of sheep infected with SPPV [42]. In our study, all three taken lymph nodes sometimes scored positive and sometimes scored negative with higher Cq-values when compared to the results of Hajjou et al. (Table 4 and Table 5). For goats infected with GTPV [25] and cattle infected with LSDV [33], a clear trend for the cervical lymph node could be observed which could not be confirmed for SPPV in sheep during the present study. However, according to our data, combined sampling of cervical, mediastinal and mesenterial lymph nodes as well as lung tissue provides a robust sampling strategy for post-mortem detection of SPPV infections in sheep. 

Due to early removal of six out of eight animals in each group because of animal welfare reasons, serological examination is suboptimal. Most animals had to be removed from the trial too early for detection of serological response. However, a few observations could be made and all animals that survived until the end of the study showed positive results in both tests, except i-c sheep IND-IC-16 (Figure 5). The used commercially available DA ELISA scored positive for the first time at 7 dpi, followed by 10 dpi and 12 dpi. Interestingly, in some cases, the DA ELISA turned to negative again at later time points (Figure 5A,C). This phenomenon could be explained by the design of the ELISA. A similar observation, probably related to the switch of Ig class, was observed for a DA ELISA detecting bluetongue disease virus antibodies [44]. Furthermore, since the Capripox DA ELISA was developed and validated for detection of antibodies against LSDV in cattle, sensitivity against SPPV-specific antibodies might be relatively low. In contrast, the SNT was positive around 14 dpi to 21 dpi for the first time (Figure 5B,D). However, this test is highly specific and sensitive and, next to the virus neutralization assay, recommended by the OIE [15]. The observed discrepancy between ELISA and SNT might be due to the type of antibodies. Whereas the ELISA detects different types of antibodies, the SNT provides insight into presence or absence of neutralizing antibodies only.

### 4.3. Establishment of a Challenge Model for SPPV Infections

Neither differences in the severity of clinical signs (Figure 1) nor marked variances in viral replication (Figure 3 and Figure 4) were detected between i.v. inoculation and i.n. inoculation in the group inoculated with SPPV-“India/2013/Surankote”, displaying that both tested infection routes are appropriate. However, i.n. inoculation carries the risk of contamination directly after experimental infection. Nevertheless, natural infection is represented by i.n. infection more closely, which is a notable advantage in a challenge model. In contrast, i.v. inoculation reduces the risk of transfer of challenge virus from vaccinated and challenged animals towards naïve contact animals right after inoculation, but does not reflect natural infection as true as i.n. inoculation does. In addition, pathogenesis of both tested virus strains was highly comparable, and no significant differences were observed during the presented study. Animals severely affected showed clinical signs characteristic for SPPV infections (Figure 1 and Figure 2), and viremia (Figure 3), viral shedding (Figure 4) and seroconversion (Figure 5) could be detected. In addition, transmission of virus from infected to naïve animals via direct contact or aerosols could be seen (Figure 1, Figure 3 and Figure 5). However, experimental infection with SPPV-“India/2013/Surankote” led to early euthanasia of all three i.v. inoculated sheep (Table 1), whereas two out of three sheep i.v. inoculated with SPPV-“Egypt/2018” recovered completely from the infection (Table 2). In addition, i.n. inoculated animals of the SPPV-“India/2013/Surankote” group had to be removed from the trial generally few days earlier than i.n. inoculated sheep of the SPPV-“Egypt/2018” group (Table 1 and Table 2). Therefore, although both SPPV-“India/2013/Surankote” and SPPV-“Egypt/2018” display very suitable candidates for challenge infections due to the observed pathology, SPPV-“India/2013/Surankote” seems to be the more efficient challenge virus strain, e.g., in future vaccine studies.

## 5. Conclusions

During our study, both tested SPPV isolates were able to induce severe clinical disease in sheep, and intravenous as well as intranasal inoculation proofed to be efficient routes for experimental infection of sheep with SPPV. In addition, in-contact animals of both groups got infected via direct contact or aerosol transmission. Nevertheless, intranasal inoculation of SPPV-“India/2013/Surankote” turned out to be the more natural and more efficient challenge model for future vaccine studies. 

## Figures and Tables

**Figure 1 microorganisms-08-02001-f001:**
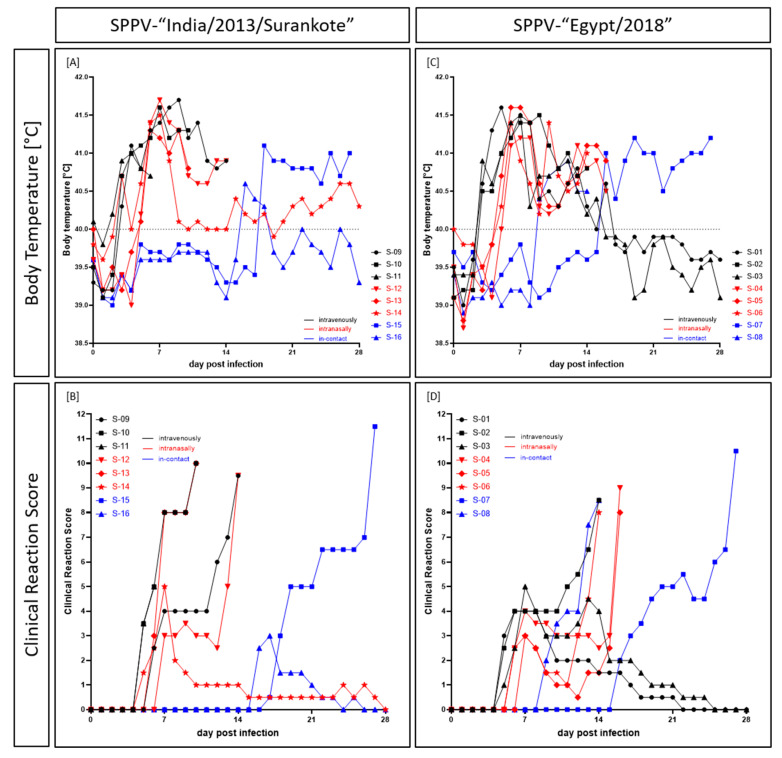
Body temperature and clinical signs observed after experimental infection of sheep with either (**A**,**B**) SPPV-“India/2013/Surankote” or (**C**,**D**) SPPV-“Egypt/2018”. Animals were inoculated via three different routes: intravenous infection (black), intranasal infection (red), and transmission between infected and naïve animals (blue).

**Figure 2 microorganisms-08-02001-f002:**
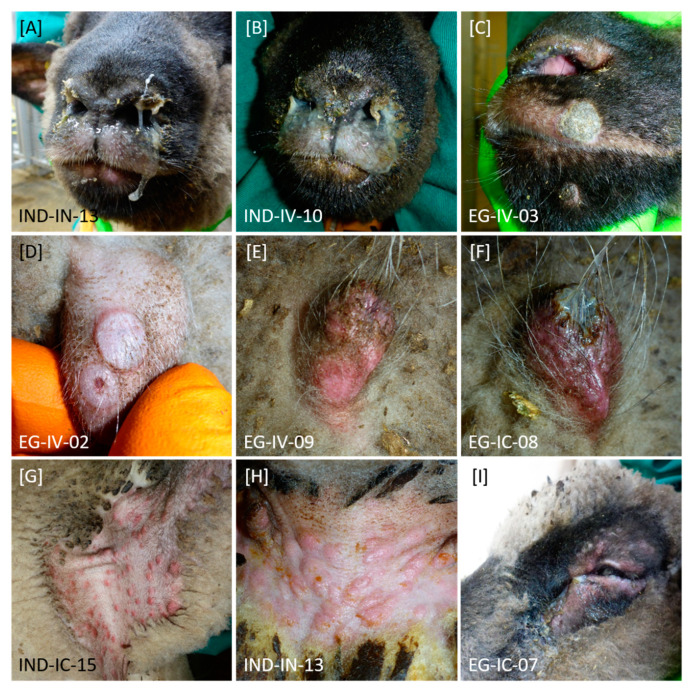
Clinical signs observed after experimental infection of sheep with SPPV-“India/2013/Surankote” and SPPV-“Egypt/2018”, respectively. Differences in type of clinical signs between both groups could not be detected. (**A**–**C**) In diseased animals, nasal discharge with different severity could be observed. (**C**–**I)** In addition, sheep showed pox lesions characteristic for SPPV infections. Here, (**D**–**F**) especially prepuce and other regions without wool like (**G**) axilla and (**H**) groin were affected. Furthermore, these lesions could be observed (**C**) around the nose and mouth as well as (**I**) around the eyes. (**I**) Few animals additionally displayed ocular discharge.

**Figure 3 microorganisms-08-02001-f003:**
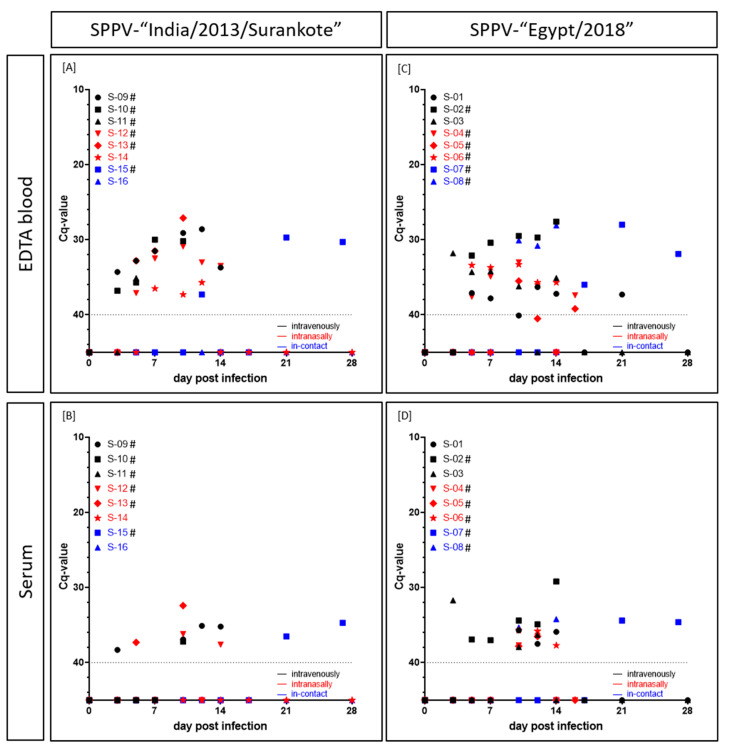
Cell-associated as well as cell-free viremia (Cq-values) of sheep experimentally infected with either (**A**,**B**) SPPV-“India/2013/Surankote” or (**C**,**D**) SPPV-“Egypt/2018”. EDTA blood and serum samples were examined using the pan-capripox real-time qPCR. Thereby, (**A**,**C**) cell-associated and (**B**,**D**) cell-free viremia was analyzed. Animals were inoculated via three different routes: intravenous infection (black), intranasal infection (red) and transmission between infected and naïve animals (blue). Cut-off was set at Cq 40.0. # displays animals that were euthanized before the end of the study due to ethical reasons.

**Figure 4 microorganisms-08-02001-f004:**
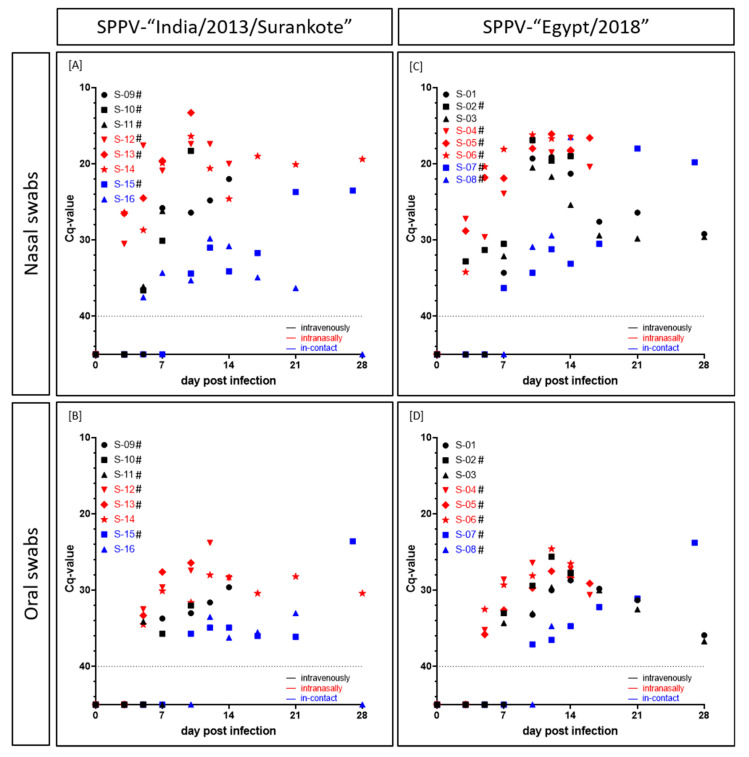
Viral shedding (Cq-values) of sheep experimentally infected with either (**A**,**B**) SPPV-“India/2013/Surankote” or (**C**,**D**) SPPV-“Egypt/2018”. (**A**,**C**) Nasal and (**B**,**D**) oral swab samples were tested using the pan-capripox real-time qPCR. Thereby, shedding of viral DNA was analyzed. Animals were inoculated via three different routes: intravenous infection (black), intranasal infection (red) and transmission between infected and naïve animals (blue). Cut-off was set at Cq 40.0. # displays animals that were euthanized before the end of the study due to ethical reasons.

**Figure 5 microorganisms-08-02001-f005:**
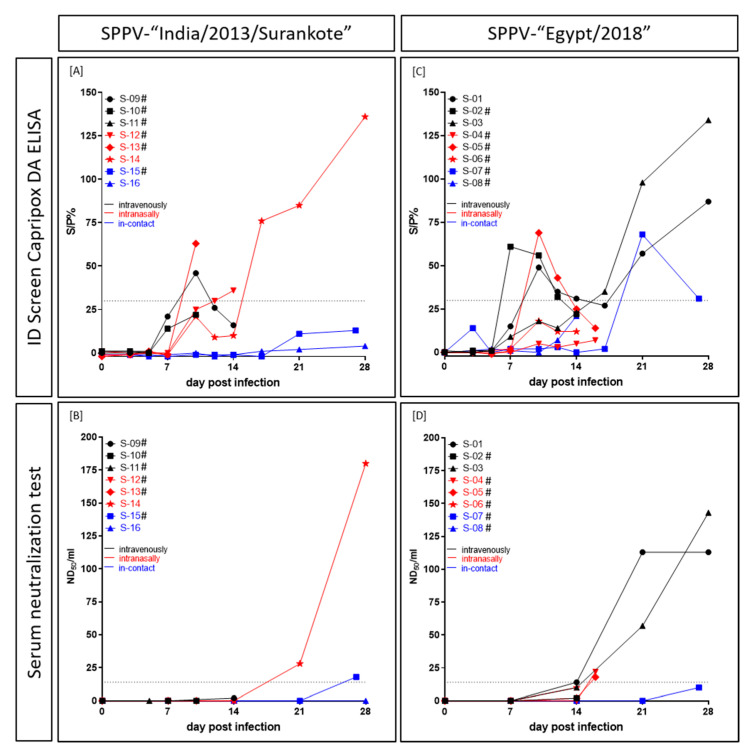
Serological response of sheep experimentally infected with (**A**,**B**) SPPV-“India/2013/Surankote” and (**C**,**D**) SPPV-“Egypt/2018”, respectively. For serological analyses, (**A**,**C**) the DA ELISA and (**B**,**D**) the SNT were performed. In the ELISA, an S/P% ratio ≥30 was defined positive, and in the SNT, ND_50_/_mL_ ≥14 was set positive. # displays sheep that were euthanized due to ethical reasons before the end of the study.

**Figure 6 microorganisms-08-02001-f006:**
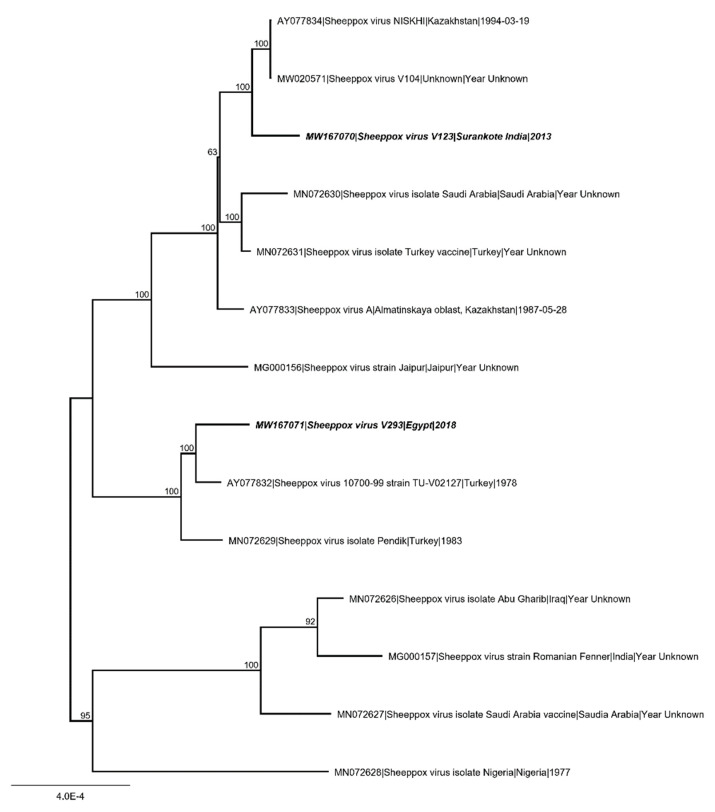
Phylogenetic analysis of Sheeppox virus (SPPV). Complete SPPV genome sequences were aligned using MAFFT. Subsequently, maximum likelihood analysis using RAxML, including 1000 bootstrap replicates was performed. Scale bars indicate nucleotide substitutions per site. Bold typing marks the SPPV isolates analyzed in the present study.

**Table 1 microorganisms-08-02001-t001:** Summarized length of protocol for sheep inoculated with SPPV-“India/2013/Surankote”.

**Animal ID**	IND-IV-09	IND-IV-10	IND-IV-11	IND-IN-12	IND-IN-13	IND-IN-14	IND-IC-15	IND-IC-16
**Days Until Euthanasia**	14 dpi	10 dpi	7 dpi	14 dpi	10 dpi	recovered	27 dpi	recovered

**Table 2 microorganisms-08-02001-t002:** Summarized length of protocol for sheep inoculated with SPPV-“Egypt/2018”.

**Animal ID**	EG-IV-01	EG-IV-02	EG-IV-03	EG-IN-04	EG-IN-05	EG-IN-06	EG-IC-07	EG-IC-08
**Days Until Euthanasia**	recovered	14 dpi	recovered	16 dpi	16 dpi	14 dpi	27 dpi	14 dpi

**Table 3 microorganisms-08-02001-t003:** Modified clinical score system for examination and categorization of SPPV infections in sheep.

Evaluation Group	Symptomatic	Score
General condition	Unremarkable	0
	Reduced activity	1
	Reduced activity, depressive	2
	Increased resting, clearly reduced activity	3
	Apathy and/or lateral position, animal does not react to stimuli	Abandonment
Feed and water intake	Unremarkable	0
	Reduced feed or water intake, increased salivation	1
	Clearly reduced feed or water intake, slight weight loss	2
	No feed or water intake (1 day), marked weight loss	3
	No feed or water intake (2 day), weight loss >10% compared to the day of experimental infection	Abandonment
Respiratory tract	Unremarkable	0
	Slightly difficult breathing	1
	Marked difficult breathing, breathing sound, slight abdominal breathing	2
	Marked difficult breathing, clear breathing sound, moderate abdominal breathing	3
	Strong difficult breathing, strong breathing sound, strong abdominal breathing	Abandonment
Nasal discharge	Not observed	0
	Slight to moderate nasal discharge, nose not/only slightly sticky	1
	Strong nasal discharge, nose clearly sticky	2
Skin changes	Unremarkable	0
	Sporadic skin nodules/lesions/papules	1
	Many skin nodules/lesions/papules	2
	Generalized skin lesions/nodules/papules	3
	Multiple abscesses or phlegmons	Abandonment
Additional clinical signs	Lameness	1
	Mild swelling of cervical lymph node	1
	Strong swelling of cervical lymph node	2

**Table 4 microorganisms-08-02001-t004:** Viral genome load (Cq-values) of different organ samples taken during necropsy after experimental infection of sheep with SPPV-“India/2013/Surankote”.

SPPV-“India/2013/Surankote”	Capri-p32-Taq-FAM							
Organ Sample	Intravenously			Intranasally			In-Contact	
	IND-IV-09	IND-IV-10	IND-IV-11	IND-IN-12	IND-IN-13	IND-IN-14	IND-IC-15	IND-IC-16
cervical lymph node	31.0	33.0	32.4	34.7	25.9	no Cq	30.4	no Cq
mediastinal lymph node	no Cq	33.0	31.0	no Cq	no Cq	no Cq	no Cq	no Cq
mesenterial lymph node	no Cq	no Cq	no Cq	no Cq	no Cq	no Cq	35.6	no Cq
liver	no Cq	no Cq	34.1	no Cq	36.4	no Cq	35.7	no Cq
spleen	no Cq	no Cq	36.4	no Cq	34.3	no Cq	no Cq	no Cq
lung	34.3	26.8	22.8	no Cq	33.1	no Cq	37.0	no Cq
**Additional Samples**								
coagulated heart blood	-	-	29.4	-	-	-	-	-
lung fluid	-	-	31.4	-	-	-	-	-
**Location of Skin Sample**								
breast	-	29.5	-	-	28.5	-	-	-
	-	19.4	-	-	29.5	-	-	-
	-	28.9	-	-	23.1	-	-	-
	-	20.4	-	-	28.3	-	-	-
axilla	16.6	28.4	21.2	36.3	27.4	-	25.9	-
	29.7	25.0	27.9	36.5	23.9	-	23.3	-
	27.3	30.7	30.4	36.3	24.2	-	24.5	-
	29.2	20.9	24.0	37.2	23.4	-	28.1	-
back	-	-	-	-	23.3	-	-	-
	-	-	-	-	26.2	-	-	-
	-	-	-	-	17.4	-	-	-
	-	-	-	-	16.9	-	-	-
hind leg	-	32.8	-	-	21.3	-	21.0	-
	-	31.7	-	-	18.5	-	19.7	-
	-	30.5	-	-	18.4	-	20.5	-
	-	32.0	-	-	18.1	-	19.3	-
tail	-	-	-	-	-	-	19.5	-
scrotum	-	20.1	-	-	25.9	-	21.0	-
	-	-	-	-	27.8	-	19.7	-
prepuce	-	18.4	-	-	17.6	-	18.9	-
	-	18.2	-	-	-	-	16.7	-
pox-like lesion lung	35.7	-	-	-	-	-	21.9	-
	35.4	-	-	-	-	-	33.8	-

- Displays sample not taken.

**Table 5 microorganisms-08-02001-t005:** Viral genome load (Cq-values) of different organ samples taken during necropsy after experimental infection of sheep with SPPV-“Egypt/2018”.

SPPV-“Egypt/2018”	Capri-p32-Taq-FAM							
Organ Sample	Intravenously			Intranasally			In-Contact	
	EG-IV-01	EG-IV-02	EG-IV-03	EG-IN-04	EG-IN-05	EG-IN-06	EG-IC-07	EG-IC-08
cervical lymph node	no Cq	32.0	no Cq	32.2	no Cq	no Cq	27.0	28.8
mediastinal lymph node	36.5	no Cq	no Cq	37.2	no Cq	no Cq	33.2	39.3
mesenterial lymph node	no Cq	36.7	no Cq	no Cq	35.7	no Cq	37.4	no Cq
liver	no Cq	no Cq	no Cq	no Cq	no Cq	no Cq	no Cq	no Cq
spleen	37.7	31.9	37.4	39.1	no Cq	38.1	39.1	36.8
lung	no Cq	30.4	37.4	33.8	33.2	34.3	35.2	33.6
**Additional Samples**								
brain	-	-	-	-	-	-	-	no Cq
	-	-	-	-	-	-	-	no Cq
	-	-	-	-	-	-	-	no Cq
nasal septum	-	-	-	21.3	28.1	-	-	-
**Location of Skin Sample**						-	-	-
ear	-	-	-	-	16.7	-	-	-
	-	-	-	-	14.7	-	-	-
	-	-	-	-	16.0	-	-	-
foreleg	-	-	-	-	-	-	18.0	-
	-	-	-	-	-	-	20.2	-
axilla	-	22.2	-	-	-	-	-	23.8
	-	21.0	-	-	-	-	-	22.4
	-	22.0	-	-	-	-	-	28.5
	-	18.8	-	-	-	-	-	-
hind leg	-	-	-	-	-	-	19.8	-
tail	-	-	-	-	-	-	18.0	-
	-	-	-	-	-	-	19.0	-
	-	-	-	-	-	-	20.1	-
prepuce	-	16.3	-	22.3	-	-	20.5	18.2
	-	14.7	-	19.9	-	-	16.4	-
pox-like lesion lung	-	-	-	26.5	32.9	-	-	-
	-	-	-	26.2	-	-	-	-
	-	-	-	27.7	-	-	-	-
healed skin lesion	28.1	-	23.5	-	24.1	-	-	-
	27.8	-	19.8	-	-	-	-	-
	32.1	-	24.6	-	-	-	-	-
	28.4	-	30.3	-	-	-	-	-
	25.7	-	-	-	-	-	-	-
encrusted skin lesion	-	-	13.8	-	-	-	-	-
	-	-	15.5	-	-	-	-	-
	-	-	15.9	-	-	-	-	-
	-	-	17.8	-	-	-	-	-
	-	-	14.1	-	-	-	-	-

- Displays sample not taken.

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
