# Peer review of "Establishment of a Challenge Model for Sheeppox Virus Infection"

_microorganisms, 2020, doi:10.3390/microorganisms8122001_

Round 1
Reviewer 1 Report
The authors established challenge models of SPPV, which is an important pathogen in the veterinary field. Such large animals are difficult to handle and conduct infectious experiments. The results clearly show the reliability of this challenge model. This manuscript is worth publishing, but there are several minor issues before publication.
- In Fig. 1, it is difficult to recognize the differences between dark gray and right gray. I recommend changing figure format (e.g. using color or broken line).
- In the Materials and Methods section, several important information for reproducibility is missing, such as ways of body temperature measurement, preparation of inoculum (directly used virus-cell-suspension?), culture media of SFT-R cells, and virus titration.
- Please describe the humane endpoint criteria in the Materials and Methods section.
- It is necessary to describe how many nucleotides were used for the sequence alignment and phylogenetic analysis.
- The animal ID should be listed in Fig. 2.
- Line 233: Please describe why S-11 had to be removed from the trial. Is it an accidental issue?
- In Fig. 2/3 and Table 2/3, the authors showed Cq value for showing the presence of the viral genome in the samples. Cq values are correlated to viral genome copy numbers but do not represent viral load. Additionally, the authors showed only Cq for organ samples, which is not standardized. For these figures and tables, viral genome copy number per milliliter or per glam should be shown for quantitative analysis.
- In Fig. 5, the Y-axis of ND50 should be presented on a log scale (commonly log2).
Reviewer 2 Report
The manuscript entitled "Establishment of a challenge model for sheeppox virus infection" describe the comparative course of infection of an egyptian and an indian strain of SPPV inoculated through nasal, iv and in-contact infection route. The manuscript is extremly descriptive and provide an interesting description of the pathology induce by both strains, however due to the low number of animals in each group, a general tendency for each route/strain is difficult to interprete.
Some points should be adressed to increase manuscript strength :
Lane 142 typo : Montepellier>Montpellier
Lane 177 -413 : the descriptive style is dense, renaming of animals with prefixe that indicate strain/ route (s01>IVEg-01, s09>IVIn-01) might help reader to proprerly analyse results
Lane 206 : a small table that summarize the length of protocol for each animal (inoculation to euthanasia) would help to compare pathogenicity (while Kaplan Meier representation would have being better but not feasible due to the low number of animals per group)
Lane 268-269 : Indicating viremia type would be of inetrest for clarity : EDTA Blood(cell associated viremia), Serum (Cell free viremia)
Lane 352 : it is indicated that 6 animals (SPPV India) were positive by qPCR in lungs while only 5 have Cq values in table
Figure 5 : negative controls of uninfected sheep have to be added in ELISA and seroneutralisation assay.
Discussion Lane 525/532: general conclusion of the author is that both strains and route leads to similar pathogenesis/replication. The variation between animals within each route /strain is high and cannot lead to such conclusion even more as IV induce euthanisa in 3/3 at days 7/10/14 in one group and only one is euthanised at day14 and two complete the protocol up to day 28 in the other one.
Round 2
Reviewer 2 Report
Dear authors thanks for taking into account my comments, best regards